# Shade provision and its influence on water intake and drinking behaviour of Nellore cattle in feedlot in a tropical environment

M. Jordana Rivero[1]*, Julio C. Pascale Palhares[2], Taisla Inara Novelli[3],
Luciane Silva Martello[3], Simón Pérez-Márquez[1], Andrew S. Cooke[4]

1 Net Zero and Resilient Farming, Rothamsted Research, North Wyke, Okehampton, Devon, United Kingdom, 2 Emprapa Pecuaria Sudeste, São Carlos, Brazil, 3 Faculdade de Zootecnia e Engenharia de Alimentos, Universidade de São Paulo, Pirassununga, Brazil, 4 Department of Life Sciences, School of Natural Sciences, College of Health and Science, University of Lincoln, Lincoln, United Kingdom

* jordana.rivero.visiting@rothamsted.ac.uk

## Abstract

Heat stress is a significant challenge in tropical beef production systems, affecting feed intake, water intake, and overall animal welfare. This study aimed to evaluate the impact of shade provision on the water intake and drinking behaviour of Nellore steers (*Bos indicus*) in a tropical feedlot environment. A total of 47 steers (~450 kg body weight) were allocated into two groups: one with access to shade (+S) and another without (-S). Individual water intake, drinking behaviour (e.g., frequency, daily patters), and animal performance were monitored over 83 days using automated recording systems. Results showed that -S steers consumed 8% more water per day (p < 0.001), made more frequent visits to the water trough (p < 0.001), but drank less per visit (p < 0.001) and overall spend 39% more time per day drinking (p < 0.001) compared to the +S steers. Despite these differences in drinking behaviour, average daily gain and feed intake did not differ between groups (p > 0.05). Environmental factors like temperature, humidity, and solar radiation affected water intake in both groups. Higher air temperatures increased water intake by boosting drinking frequency, while higher relative humidity reduced water intake by decreasing visit frequency. Shade provision reduced water demand per unit of body weight gain, improving water-use efficiency. These findings suggest that while shade may not directly enhance body weight gain, it can optimise drinking behaviour, reduce water intake, and improve animal welfare in tropical beef production systems.

## Introduction

The rise in global average temperatures poses a major threat to livestock production [1]. Heat stress in cattle causes decreases in feed intake, growth, and efficiency, and threats animal health and welfare. In extreme cases, heat stress can cause death

**Data availability statement:** The datasets are available at Mendeley Data, DOI: 10.17632/cctgpznjrc.1 (hyperlink: https://data.mendeley.com/datasets/cctgpznjrc/1).

**Funding:** This research was partially funded by the Alliance for Sustainable Agriculture (ASA) Partnership award "Measuring Sustainability Metrics for Ruminant Livestock Production Systems in Brazil and UK for a global assessment" funded by the Newton Fund. Rothamsted Research receives strategic funding from the Biotechnological and Biological Sciences Research Council (BBSRC) of the United Kingdom. Support in writing up the work was greatly received by BBSRC through the strategic program Soil to Nutrition (S2N; BBS/E/C/000I0320) and Growing Health (BB/X010953/1). This work was partially supported Coordenaçao de Aperfeiçoamento de Pessoal de Nível Superior – Brazil (CAPES) - Finance Code 001.

**Competing interests:** The authors have declared that no competing interests exist.

of vulnerable animals [2]. Feedlot cattle are susceptible to heat stress, which can negatively impact their performance and overall wellbeing. This issue is particularly prevalent in tropical climates, where high temperatures can lead to reduced feed intake and increased respiratory rates due to elevated thermal loads [3,4]. Nearly half of the world's beef production comes from tropical and subtropical regions, and the implementation of best management practices for sustainable resource management is of global significance [5].

Cattle use evaporative cooling to dissipate heat load. However, this process increases their need to consume water to maintain homeostasis [6], therefore increasing the demand for water of the production systems. Shade reduces heat load and the need for evaporative cooling and water replenishment [7]. Artificial shade is in the forefront of environmental modifications to mitigate the negative impacts of heat stress and to improve welfare of beef cattle [8], particularly in tropical environments [9]. With increasing heat stress, cattle exhibit more changes in physiology and behavioural strategies related to heat dissipation, leading to greater competition for shade and water [10]. Therefore, heat stress affects not only water intake (WI) and access to shade but also drinking behaviour (e.g., number of visits to the water troughs) and daily WI patterns. For instance, cattle without access to shade spend more time near water troughs and visit them more frequently, increasing antagonistic interactions and reducing the time spent on more productive activities, such as ruminating or resting, compared to animals with shade access [10]. Factors influencing drinking behaviour include water source and availability, water quality, and environmental conditions. Drinking behaviour can, in turn, impact animal performance and WI as well as animal welfare; understanding the relationships among these factors is essential for designing water systems for grazing and feedlot cattle [11]. However, there is little agreement in the literature regarding the main drivers of WI and, consequently, drinking behaviour. It was hypothesised that access to shade would reduce WI under tropical conditions and modify the drinking behaviour and daily drinking patterns of feedlot cattle, and that these changes would be driven by weather conditions. Therefore, the objective of this study was to compare the effects of access to shade versus no access to shade of *Bos indicus* steers in a feedlot located in tropical conditions on daily WI and patterns, drinking behaviour and animal performance.

## Materials and methods

The spring season experiment took place from September 2019 to December 2019 at the Experimental Feedlot of Embrapa Southeast Livestock in Sao Carlos, Sao Paulo, Brazil (21° 57′ 42″ S, 47° 50′ 28″ W, 860 m height above mean sea level). No permission was required to carry out this research since Embrapa is a research institution. All procedures adopted were approved by the Animal Use Ethics Committee of the Faculty of Animal Science and Food Engineering at USP (CEUA/FZEA), which certified the use of animals in accordance with protocol No. 5011140119. The study area has a tropical climate classified as Cwa according to Köttek et al. [12]. To characterise heat stress risk, minimum, maximum and mean dry bulb temperature (DBT), black globe temperature (BGT), relative humidity (RH, %), wind speed (WS, m/s),

solar radiation (SR, W/m²) and rainfall (mm) were automatically recorded every hour, 24 hours a day, by the Embrapa weather station. The DBT and RH were used to calculate the temperature-humidity index (THI) according to the following equation: $THI = 0.8 \times DBT + [(RH/100) \times (DBT - 14.4)] + 46.4$ [13,14].

Forty-eight 22-month-old Nellore steers (*B. indicus*), with an average body weight (BW) of 450 kg, were initially included in the study. However, one animal was removed due to a leg injury, leaving a total of 47 animals for the experiment. The remaining animals were divided into two groups: one group with shade (+S) and another group without shade (-S). After an adaptation period of 11 days, steers were housed in four collective pens (400 m² each, 20 m x 20 m) (Fig 1), with 12 steers in three pens and 11 steers in the fourth pen, for a total of 83 days. Each of the four groups was randomly allocated to one of the four pens. Their BW were measured at the beginning of the adaptation period, then after 21 d, and at the end of the experimental period (just before being sent to the abattoir). The BW at the beginning of the experimental period was estimated assuming a linear growth between the first (beginning of acclimatation period) and the second (10 days after the experimental period started) weighing events.

During the experiment, the animals had unrestricted access to water and a total mixed ration. Feed delivery was ad libitum and adjusted daily to minimise refusals for the following day. The water used in the study was sourced from a well. Meals were provided at 07:00, 11:00, 14:00, and 16:00 hours daily. The diet composition included sugarcane bagasse, soybean, dry corn grain, and a mineral mix (Table 1).

The monitoring of individual WI was carried out using the Intergado™ System (Intergado Ltd., Contagem, Minas Gerais, Brazil). This system employs radio-frequency identification (RFID) technology to track individual cattle and water flow meters installed in the troughs to precisely measure the volume of water consumed during each drinking event. The water troughs, designed and sized by Intergado Ltd., were tailored to the drinking behaviour of beef cattle, ensuring sufficient access and capacity in accordance with the manufacturer's specifications. One water trough was placed in the sunny part of each pen.

Feed intake was measured using the GrowSafe™ system (GrowSafe Ltd., Calgary, Alberta, Canada). This system employs RFID technology to monitor individual cattle and records feed consumption through load cells integrated into the feeding bunks, which continuously weigh the feed to determine intake during each feeding event. Two GrowSafe™ units were available per pen, located under a small roof (Figs 1–4). Periodic feed samples (~weekly, i.e., every time a new

**Table 1. Feeds composition and nutritional value of the diet fed to Nellore steers with and without access to shade finished in feedlot during spring in a tropical environment.**

| Feed ingredient | Proportion of the total mixed diet (g/kg DM) |
|---|---|
| Sugarcane bagasse | 144 |
| Soybean | 58 |
| Dry corn grain | 739 |
| Mineral mix | 59 |
| **Nutritional component** | **% of DM (mean ± SEM)** |
| Crude protein | 16.6 ± 2.44 |
| Neutral detergent fibre | 24.2 ± 2.88 |
| Acid detergent fibre | 13.3 ± 3.04 |
| Ether extract | 2.5 ± 0.56 |
| Lignin | 2.0 ± 0.56 |
| Ash | 6.36 ± 0.52 |
| Neutral detergent insoluble nitrogen | 0.28 ± 0.04 |
| Acid detergent insoluble nitrogen | 0.24 ± 0.02 |
| In vitro digestibility (%) | 78.9 ± 3.78 |

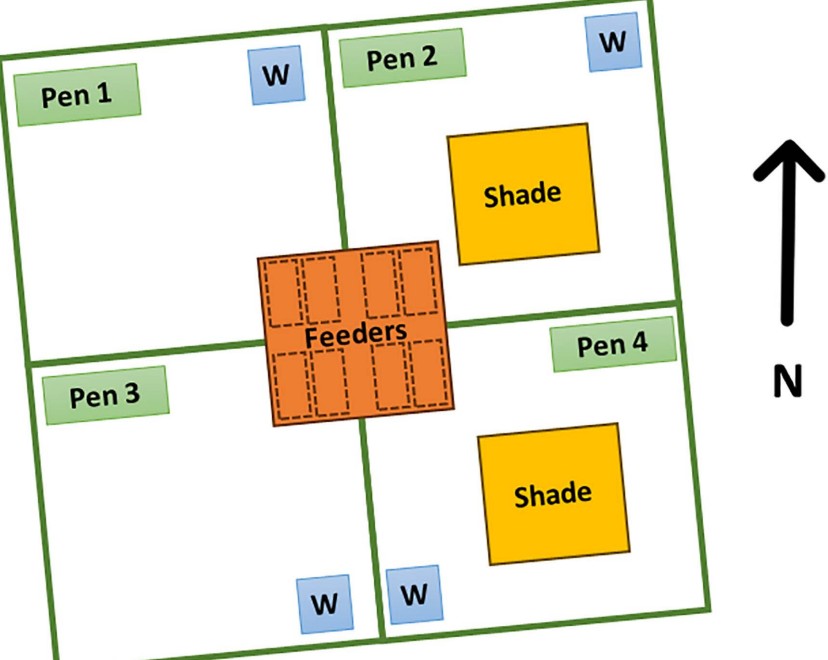

**Fig 1. Layout of the locations of the four pens (two shaded, two unshaded), the water through ("W"), the shaded automatic feeders (two per pen) and the shade structures in a feedlot system for Nellore steers.**

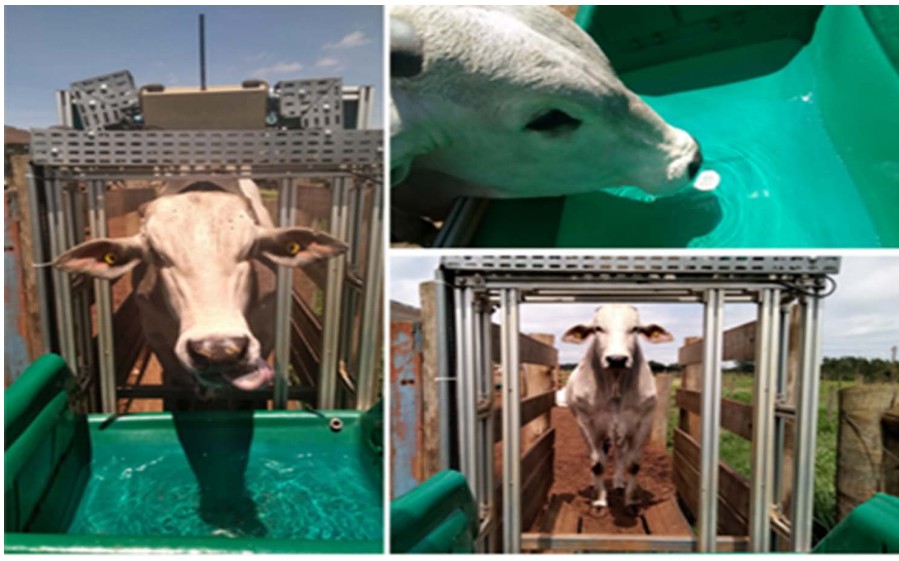

**Fig 2. Intergado™ System and steer drinking water.**

batch of sugarcane bagasse was introduced) were collected from the feed troughs and dried in a forced-air oven at 65 °C for 72 hours to determine the dry matter (DM) content. The resulting value (mean ± standard deviation: 84.0 ± 3.24%) was used for calculating the daily DM intake (DMI) of each animal. Dried samples were analysed for nutritional composition (DM basis); the lab values are presented in Table 1.

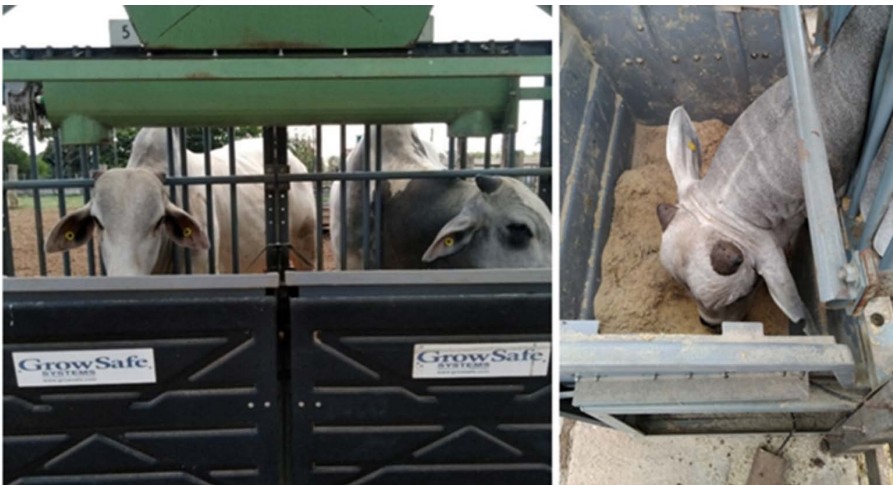

**Fig 3. GrowSafe™ system and steer eating.**

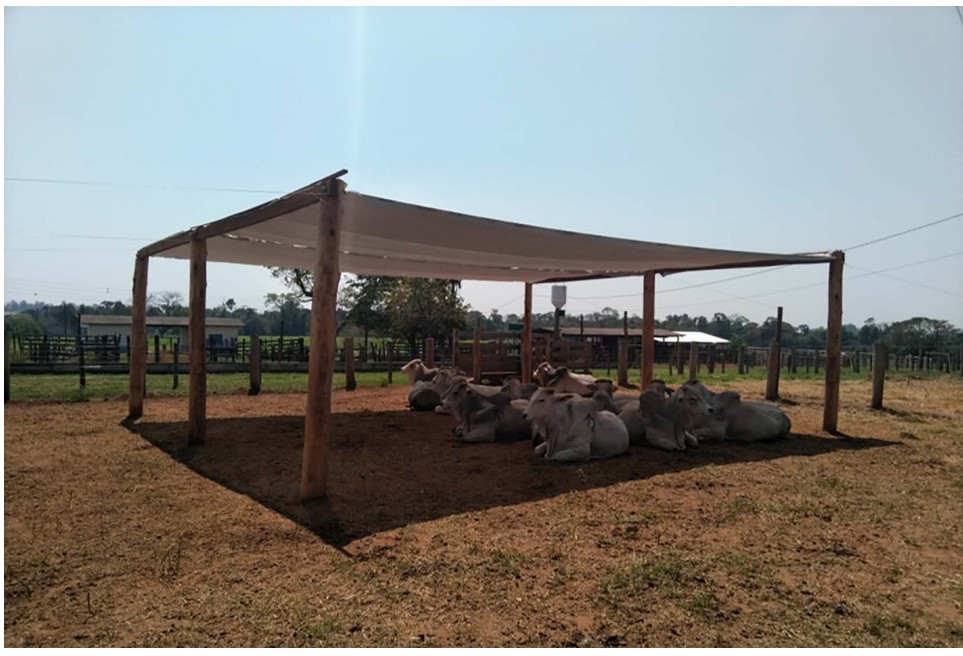

**Fig 4. Structure used to provide shade.**

To provide shade for the animals, an artificial structure was constructed with dimensions of 9 m × 8 m (i.e., 72 m²), oriented in the East-West direction (Fig 4). The shade material used was a thermo-reflective aluminised mesh that reflects ultraviolet and infrared rays. The shade structure was designed based on reference values of 6 m² per animal and a height of 3 m [2,15]. The manufacturer specifications ensure 78–83% shade and 32% diffused light transmission.

The BW recorded was used to calculate the individual average daily gain (ADG, kg/d) during the experimental period, whereas the datasets from the Intergado System allowed to calculate the variables: individual WI, measured daily (L/d) and over the total period (83-d), water consumed per visit to the water trough (L/visit), number of visits to the water

troughs (with consumption) per day, time spent drinking per day (s/d), visit (to the water trough) duration (s/visit) and WI rate (L/min). Additionally, WI was expressed per kg ADG (L/kg ADG), in litres consumed as a percentage of BW, and per kg DMI (L/kg DM). The mean, median, coefficient of variation (CV, %) and standard error of the mean across days was calculated using the average value calculated across the animals of each group.

The 47 individual animals were considered the experimental units and the groups compared were +S and -S. Data on animal performance and WI variables were analysed using paired Wilcox signed rank tests that compared the mean values for a variable (e.g., WI) of each group on each day. A zero-inflated Poisson model was used to assess the effects of time of day, treatment group, and their interaction on hourly WI, accounting for excess zeroes in the data. Differences between means of +S and -S were considered significant at $\alpha < 0.05$. Spearman correlation coefficients were calculated between the daily mean of weather variables and the daily mean of the response variables for each group, with a Bonferroni adjustment applied to account for multiple (78) comparisons. Data were analysed with R-Studio (2023.09.1) using the package 'stats' [16] and 'pscl' [17]. The package 'ggplot2' [18], in combination with package 'Cairo' [19], was used to create figures.

## Results

### Environmental variables

Average mean DBT across the 83-d experimental period was 22.9 °C (ranged from 16.6 and 27.5 °C), whereas that recorded inside the BGT was 25.7 °C (ranged from 16.8 to 30.3 °C) (Fig. 5). Average minimum and maximum temperature were 17.8 and 29.1 °C, respectively. Average RH was 70.8% (ranged from 25.8 to 99.5%) and the average WS was 2.7 m/s (ranged from 0.8 to 6.1 m/s), whereas average SR was 245 W/m² (ranged from 23.3 to 352 W/m²). Hence, the average THI was 69.9 (ranged from 61.9 to 78.4), with each day averaging 12 h of THI values above 70 and 4.6 h above 75. None of the days showed any 15-min value with THI above 80. The accumulated rainfall during the four 21-d periods were 39.4, 33.0, 116.4 and 99.8 mm, respectively.

### Animal performance

Initial, medium, and final BW did not differ between steers with (+S) and without (-S) access to shade (p > 0.05), averaging 477 ± 3.9 kg, 551 ± 4.8 kg, and 591 ± 5.0 kg, respectively. Consequently, ADG across the experimental period did not vary between groups, averaging 1.43 ± 0.046 kg/d. Dry matter intake did not vary between groups either, averaging 11.6 kg DM/d per animal (p = 0.660) and 2.11% BW (p = 0.276). Therefore, feed conversion ratio did not differ (p = 0.399) between groups either, averaging 8.26 ± 0.285 kg DM/kg BW gained.

### Water intake and daily drinking behaviour

Steers in the -S group consumed approximately 8% more water per day on average compared to those in +S (V = 563, p < 0.001) (Table 2). This was characterised by -S animals making nearly one additional visit to the water trough per day (V = 321, p < 0.001), spending 16% more time drinking per visit (V = 48, p < 0.001), and 39% more time drinking per day (V = 5, p < 0.001). Despite this, -S animals consumed 7% less water on each visit to the trough (V = 3344, p < 0.001) due to their 12% lower drinking rate (V = 3479, p < 0.001). The amount of water consumed per kg of ADG was greater (p = 0.025) for the -S (29.6 ± 1.21 L/kg ADG) than that of +S (25.5 ± 1.18 L/kg ADG), representing a 15% increase. Summed across the whole experimental period, -S steers consumed 338 L more (p = 0.031) than those in +S (3267 vs. 2930 L/animal). Hence, the WI:DMI ratio tended (p = 0.053) to be 9% higher in the -S than in the +S group (3.20 ± 0.107 vs 3.50 ± 0.109 L/kg DM). When added together DMI and WI, the -S group had a higher (p = 0.048) combined intake (52.6 ± 1.54 vs 48.2 ± 1.51 kg) than the +S, representing a higher (p = 0.017) percentage of the animals' BW (9.65% ± 0.278 vs 8.68 ± 0.272). DMI correlated positively with WI (r = 0.48, p < 0.001).

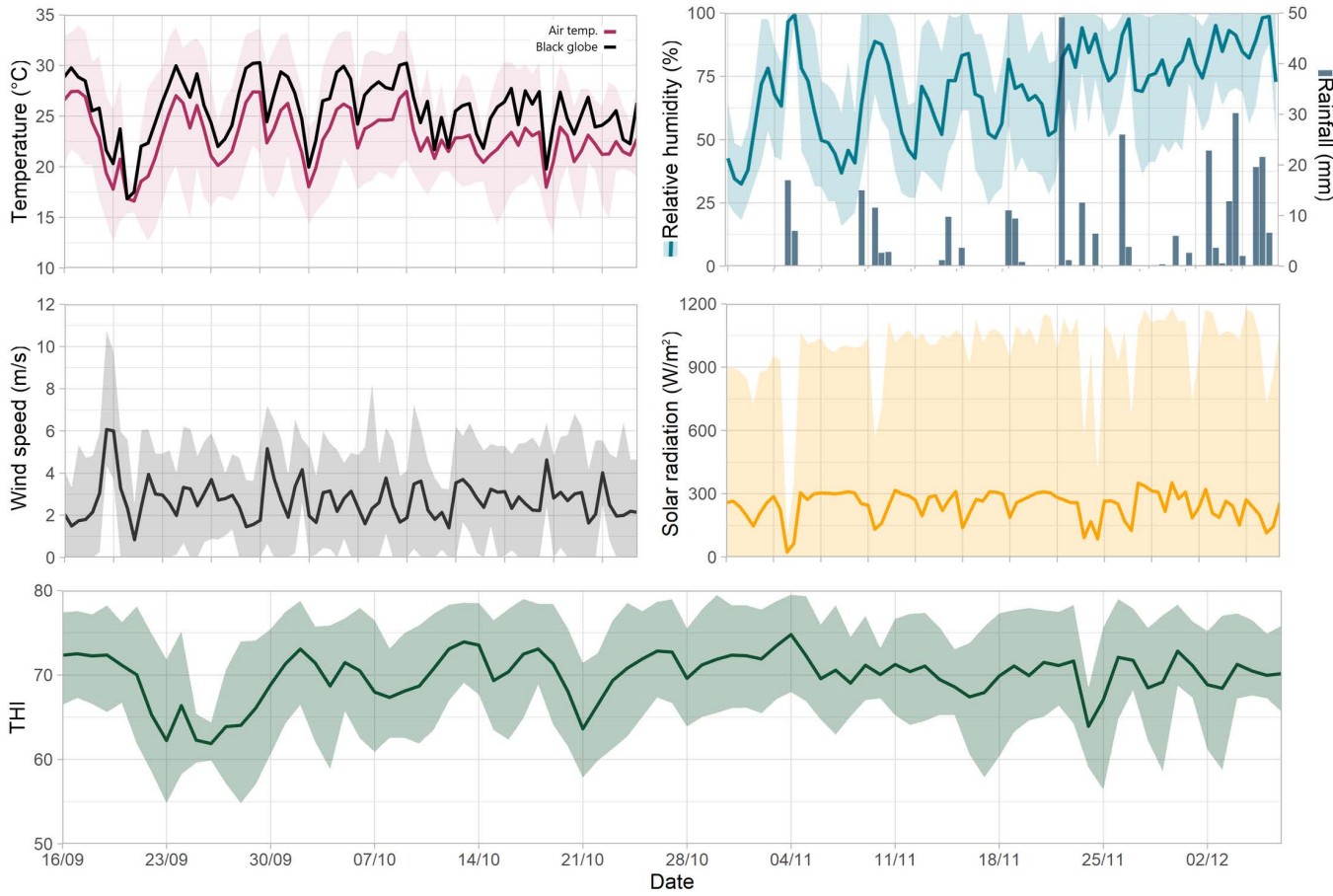

**Fig 5. Weather conditions throughout the study period.** Top left: Air temperature and black globe temperature (°C). Top right: Relative humidity (%) and rainfall (mm). Middle left: Wind speed (m/s). Middle right: solar radiation (W/m²). Bottom: Temperature humidity index (THI). Shaded areas represent 95% confidence.

**Table 2. Water intake and drinking behaviour in Nellore steers with (+S) and without (-) access to shade finished in feedlot during spring in a tropical environment.**

| Variable | Mean | | Median | | Range | | CV | | SEM | | p − value |
|---|---|---|---|---|---|---|---|---|---|---|---|
| Shade group | +S | -S | +S | -S | +S | -S | +S | -S | +S | -S | |
| Water intake (L/d) | 35.1 | 38.1 | 35.0 | 37.3 | 24.1-46.9 | 25.3-63.4 | 13.5 | 18.8 | 1.0 | 1.6 | <0.001 |
| Drinking rate (L/min) | 4.2 | 3.7 | 4.1 | 3.4 | 2.6-7.9 | 1.5-6.9 | 32.5 | 38.4 | 0.3 | 0.3 | <0.001 |
| Visits per day | 4.6 | 5.5 | 4.3 | 5.2 | 3.0-6.8 | 2.2-9.2 | 23.1 | 26.4 | 0.2 | 0.3 | <0.001 |
| Water consumed per visit (L/visit) | 8.2 | 7.5 | 7.8 | 7.0 | 5.6-11.4 | 4.7-11.6 | 20.5 | 21.6 | 0.4 | 0.4 | <0.001 |
| Time spent drinking (s/d) | 688 | 953 | 663 | 972 | 349-1136 | 317-1823 | 28.7 | 41.9 | 41.1 | 87.6 | <0.001 |
| Visit duration (s/visit) | 153 | 177 | 153 | 158 | 76-247 | 68-308 | 25.2 | 35.8 | 8.0 | 14.2 | <0.001 |

Mean, median, range, coefficient of variation (CV) and standard error of the mean (SEM) were determined from the mean values for each animal over the study period. *p*-values were calculated from Wilcox signed rank tests comparing the means of each group on each day.

Notably, across all measured WI variables, variation was greatest among cattle without shade (Table 2). Particularly, the time spent drinking and the visit duration in the -S animals showed a 5–6-fold increase between the minimum and maximum values. The drinking rate also showed a great variability in both groups.

## Impact of weather on water intake

Weather impacted drinking behaviour with comparable effects seen across both groups (Fig 6). There was a positive correlation of WI with mean BGT (+S: $r_s$=0.617, $p$<0.001; -S: $r_s$=0.542, $p$<0.001) and maximum temperature (+S: $r_s$=0.682, $p$<0.001; -S: $r_s$=0.616, $p$<0.001). This was driven by an increase in the number of visits to the water trough per day, as opposed to an increased in water consumed per visit. For both groups, WI was negatively associated with rainfall (+S: $r_s$=−0.675, $p$<0.001; -S: $r_s$=−0.577, $p$<0.001), mean RH (+S: $r_s$=−0.669, $p$<0.001; -S: $r_s$=−0.609, $p$<0.001) and maximum RH (+S: $r_s$=−0.709, $p$<0.001; -S: $r_s$=−0.574, $p$<0.001). Wind speed had no association with WI variables, however no extreme highs of WS were observed during the study period. Mean SR had a positive association with daily WI (+S: $r_s$=0.525, $p$<0.001; -S: $r_s$=0.515, $p$<0.001), whilst maximum SR was positively associated with increased water consumption per visit (+S: $r_s$=0.539, $p$<0.001; -S: $r_s$=0.549, $p$<0.001).

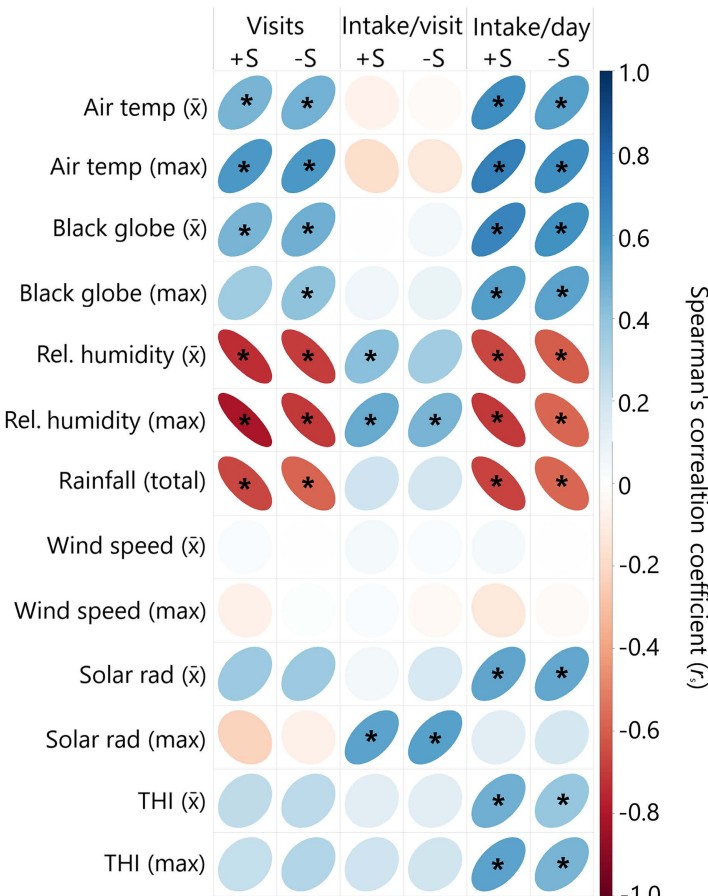

**Fig 6. Spearman's correlation matrix of weather conditions, both mean ($\bar{x}$) and maximum, against daily number of visits to the water trough, mean water intake per visit, and mean water intake per day, for both groups.** All $r_s$ and p-values can be found in Supplement A. Ellipse shape and colour represents correlation direction and strength. Cells with an asterisk (*) are statistically significant after a Bonferroni correction for multiple (78) comparisons.

## Within and between day variation in water intake

Daily drinking patterns varied between treatment groups (Fig 7). Time of day was associated with WI (z = −20.90, p < 0.001), with a general decline in WI as the day progressed. Across both groups, the number of visits to the water troughs was greatest in the mid-late afternoon (~15:00–18:00), although total WI was relatively lower during these periods compared to earlier in the day. There was an interaction effect between time of day and treatment group (z = 4.37, p < 0.001), indicating that -S steers exhibited a greater temporal variation in their drinking patterns, while +S steers drank more consistently throughout the day. Additionally, the likelihood of having zero drinking events decreased as the day progressed (z = −25.80, p < 0.001), suggesting that animals were less likely to have no drinking events later in the day.

Over the course of the study, for both groups, the number of visits per day gradually declined (Fig 8). However, the water consumed per visit increased over the same time (Fig 9), resulting in only a slight reduction in WI over the study period (Fig 10), though this trend was highly variable.

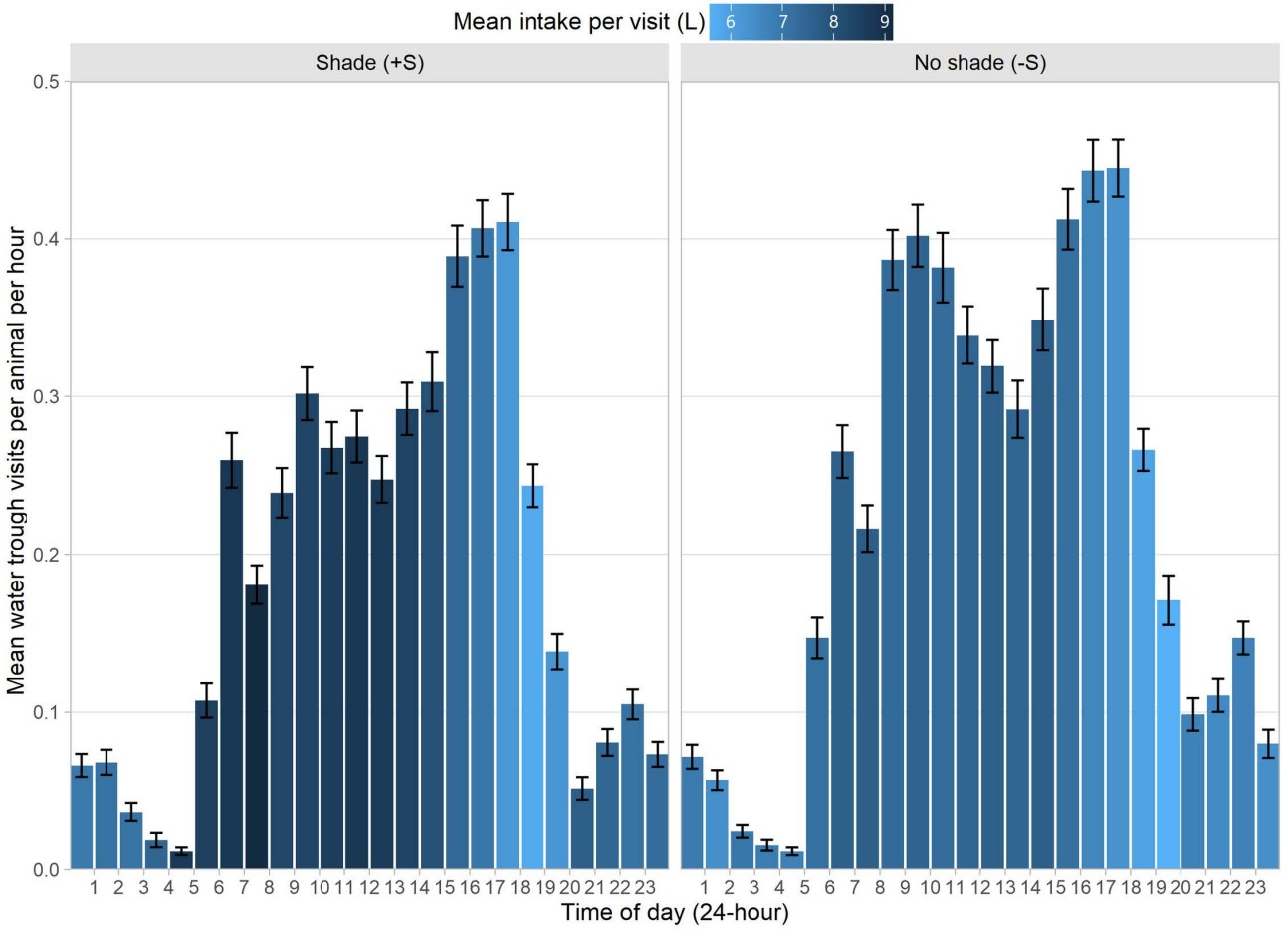

**Fig 7. Frequency of trough visits throughout the day.** Values are the mean number of visits per animal, for each hour of the day, split by group, with error bars representing standard error. Bar colour represents the mean water intake (litres) per visit for the relevant hour of the day.

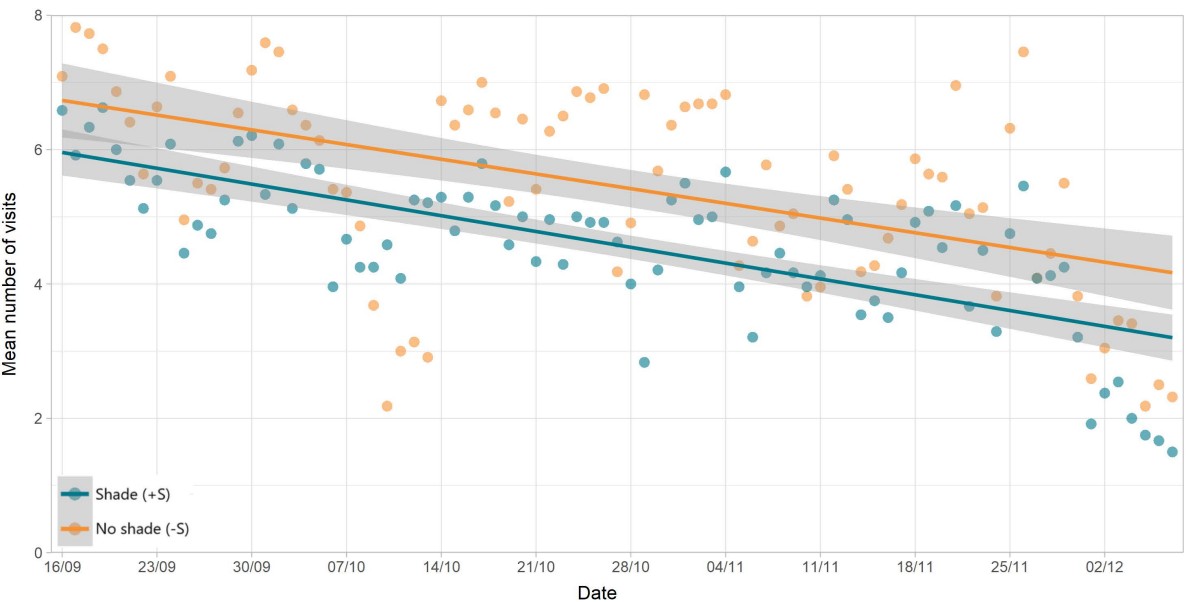

**Fig 8. Change in drinking behaviour over time: number of visits.** Mean number of visits to the water trough, by group, across the study period. Green points are animals with shade (+S) and orange are those without (-S). Shading around trend lines represent 95% confidence.

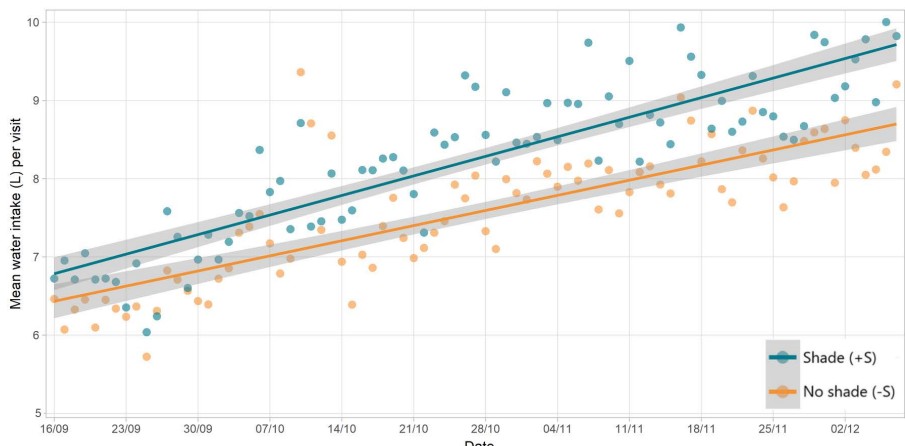

**Fig 9. Change in drinking behaviour over time: water intake per visit.** Mean intake (L) of water per visit, by group, across the study period. Green points are animals with shade (+S) and orange are those without (-S). Shading around trend lines represent 95% confidence.

## Discussion

### Water intake and shade

Drinking behaviour of Nellore steers varied based on whether the cattle had access to shade or not. Those with shade drank less frequently but drank more per visit, than those without shade. The combined effect was that cattle with shade drank less per day than those without. These results highlight how environmental modifications can impact the thermo-regulatory strategies of cattle and may inform management strategies for heat stress. Whilst these results are broadly consistent with the literature, one novel finding was that cattle with shade typically loaded with water earlier in the day,

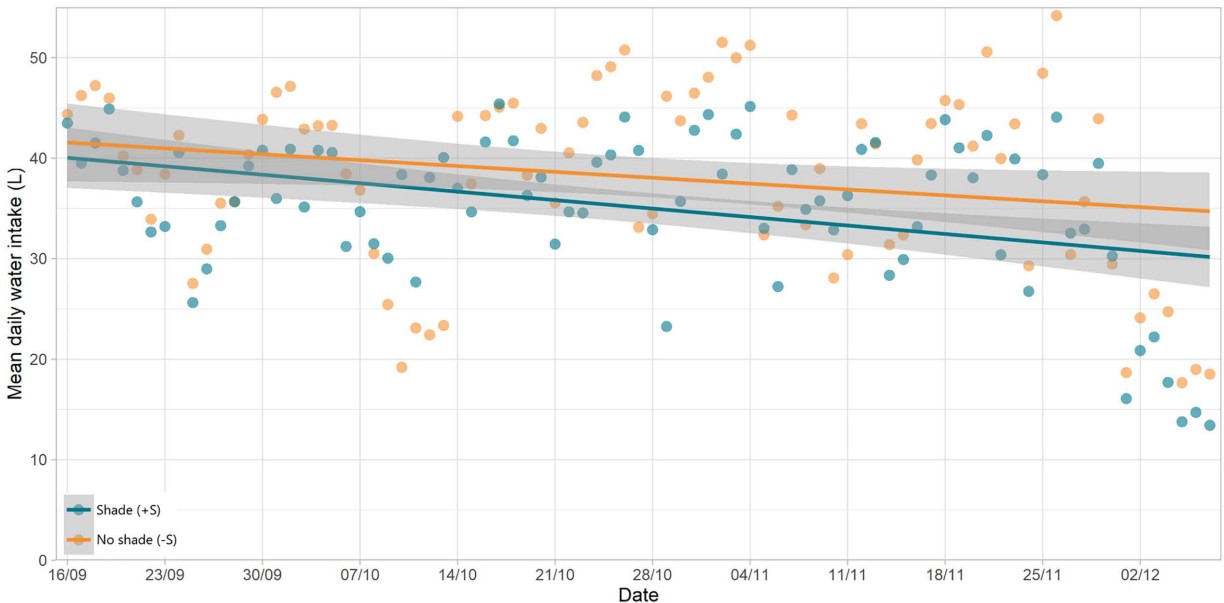

**Fig 10. Change in drinking behaviour over time: daily water intake.** Mean daily water intake per animal, by group, across the study period. Green points are animals with shade (+S) and orange are those without (-S). Shading around trend lines represent 95% confidence.

whilst those without drank at a more consistent rate. Despite being exposed to warm conditions, with the majority of days experiencing a THI above 70 (even at nighttime), and an average BGT about 3 °C above the DBT, the animals grew at a good rate, comparable to values reported for Nellore cattle in feedlots under similar conditions [20–23] and with similar DMI levels when expressed as a percentage of BW [20,22,24]. This finding aligns with other research that suggests shade may not always impact growth performance, even though it reduces WI [25,26]. The lack of differences in DMI and ADG between the groups in our study could be attributed to the fact that drinking water was sufficient for thermoregulation, allowing the animals to maintain feed intake and growth rates despite the differences in heat stress levels. Additionally, it has been reported that Nellore cattle have the ability to adjust their body metabolism to the adverse environmental conditions during the day, especially in the semi-arid tropical regions [25].

The differences in drinking behaviour may be attributed to heat load. Shade reduces heat load and the need for evaporative cooling and water replenishment [7]. Our results are consistent with the literature [7,27], indicating that shade plays a vital role in reducing water demand. Daily WI reported in our study (35.1 L/d [6.8% BW] for +S and 37.9 L/d [7.3% BW] for -S) were broadly consistent with literature on Nellore cattle; for example, Zanetti et al. [28] reported an intake of 6.5% BW for 299-kg animals. Animals in +S had two options to use their environment to thermoregulate, i.e., water ingestion or use of shaded area; however, animals in -S only had access to water. Thermoregulation through shade use is likely the primary strategy when SR significantly contributes to heat stress, reducing the need for thermoregulation by WI.

## Drinking behaviour and daily patterns

Feedlot cattle most actively drink between 0600 and 2100 h [11,29], which agrees with our findings. When cattle have access to shade they remain there during the hottest hours of the day, leaving shade only when looking for water or at the end of the day. In our study, shaded animals exhibited distinct drinking patterns, consuming more water per visit but less frequently, particularly during cooler hours of the day, which aligns with their adaptation to minimise heat stress during peak temperatures. This adaptation can be explained by the lower water search frequency observed in animals with

shade, which is also associated with the milder microclimate created by shade [30], and a reduced incidence of direct SR. The faster drinking rate of shaded animals could indicate that they prioritised efficiency, drinking more in less time. This behaviour allowed them to optimise time for feeding and resting, reducing exposure to heat, since cattle with access to shade optimise their drinking behaviour, allowing them to spend less time near water sources and more time engaged in productive activities like ruminating [10], and grazing, particularly during peak hours than cattle with no access to shade [31]. These differences in behavioural activities are larger when THI is above the bovine comfort zone over 24 h [31]. In our study, the frequency of visits in the peak hours seems not to differ between +S and -S groups, but the amount of water consumed per visit in the morning was greater in the +S; this could imply that steers with access to shade consumed most of their water outside the peak hours by drinking faster an in fewer occasions to dedicate more time to eat and laydown in the colder hours of the day.

## Environmental factors affecting drinking behavior

As the severity of heat stress increases, cattle exhibit increasingly more changes in physiological and behavioural attributes related to heat dissipation, increasing competition for shade and water to the point where the benefits outweigh the costs [10]. For instance, time spent drinking increases with THI, but the rate of increase is more than two-fold in animals without access to shade [32]. Maximum shade utilisation of cattle occurs when levels of SR exceed 800 W/m$^2$ [9]. Applying the probability distribution proposed by Maia et al. [9] and considering that, in our study, several days had SR values above 300 W/m$^2$, this would imply that around 50% of the animals would look for shade in the morning and around 75% in the afternoon. Rain could be another factor influencing animal behaviour. In our study, the rainy days had an average maximum DBT 3.2 °C lower and a BGT 2.7 °C lower than the days with no rain (see raw data files [33]). In rainy days, the animals in -S decreased they daily WI by 9.1 L/d while the ones in +S decreased it only by 6.4 L/d, while both groups decreased the number of daily visits to the water troughs by around 1 and increased the visit duration by 30 seconds (see raw data files [33]). One potential explanation for this reduced WI and visits could be due to the reduced level of overall activity in rainy days [9]. Another reason could be the milder temperatures experimented in rainy days. When comparing three days with rain and three days without rain with a similar average BGT (~21.5 °C), we found that the daily WI was similar within the -S and the +S groups (around 33.4 and 31.3 L/d, respectively) but with ~0.6 more daily visits on the days with no rain for -S (4.9 v 5.4) and +S (4.2 vs 4.9) (see raw data files [33]), which would support the idea that the rain, in addition to create milder conditions, also reduced the level of activity of the animals. When comparing the three warmest days (BGT = 30.2 °C, no rain) with the three coldest days (BGT = 18.0 °C, rain), animals in -S increased their WI by 18 L/d, while the ones in +S increased 13.6 L/d, whereas both groups increased their daily visits by around 1.2 (see raw data files [33]). This would suggest that the animals in -S increased their drinking rate in hot days in comparison with those in +S. Water temperature increases linearly with DBT (R$^2$ = 0.95) [9]; thus, warmer water in very hot days may affect drinking behaviour. However, given that the water troughs in our study had a maximum capacity of 50 L, the water renewal rate was higher than that of traditional water troughs, which resulted in animals having fresher water. Ultimately, WI (both free and in feed) is driven by multiple factors, including BW, ADG, feed characteristics, and environmental conditions, resulting in high variability. Wagner and Engle [11] reported, for finishing cattle of 544 kg (median of 551 kg in our study), a demand of 53.4–61.3 L/d (9.8–11.3% BW) depending on ambient temperature, whilst Lardner et al. [34] found yearling steers of 304 kg to require 42 L/d (13.8% BW).

We aimed to disentangle the effect of weather conditions on drinking behaviour by analysing the relationships between these variables. There is little agreement in the literature as to the main drivers of WI; whilst some studies report maximum DBT to be the main factor [28,35] others report minimum DBT or THI [6], while RH has also been reported as a major contributor to heat stress in hot climates [36]. Our results agree with the former, with maximum DBT strongly correlating to daily WI. Considering that maximum DBT values tend to increase due to global warming [7], this result is important for water planning in cattle feedlots. With regards to RH, whilst we found that is positively correlated with water

consumed per visit, the negative relationship it had with the number of visits per day was strong enough that RH had a net negative relationship with overall daily WI.

In tropical climates, rising RH limits the efficiency of evaporative cooling [37], forcing cattle to drink more water to regulate their body temperature. The low average WS recorded in our study could be classified as a light breeze [38], which may have contributed only with a small amount of convection cooling, hence the not significant association with WI or drinking behaviour in our study. This negligible convection cooling potential of the environment due to wind may have increased the need of the animals to dissipate heat through evaporate cooling [39], amplifying the impact of RH in the heat load of the animals and their drinking behaviour, hence the strong correlation between RH and WI and number of visits to the water troughs per day. The overall trend observed over time in the decrease in DBT and the increase in RH would then contribute to explain the overall trend of decreasing the number of visits to the water troughs, the increase in the amount of water consumed per visit and the decrease in the mean daily WI observed as the finishing feedlot phase progressed.

Exposure to SR can increase body temperature, which can lead to increases in daily WI to help regulate body temperature [6]. Consequently, in our study, daily WI increased as SR increased, and, interestingly, steers drank more water per visit as the maximum SR increased. Unexpectedly, THI and maximum THI were not associated with the drinking behaviour (visits per day or water consumed per visit) in our study. Souza et al. [32] found strong correlation between THI and daily WI in hot climate, with THI ranging from 74 to 86, whereas in our study average THI ranged from 61.9 to 78.4. This would suggest that the effect of the THI would be relevant in warm conditions, as shown by the significant correlation of the maximum THI on the amount of water consumed per visit. Interestingly, rain appears as a not important variable to model daily WI [11]. This contrasts with our findings since rainfall was negatively correlated with daily WI, mainly driven by the number of visits.

WI is likely circumstantial and unique to each production system. Because of this, attempts at modelling water demand have had limited success due to the complexity of its drivers, rarely being able to explain much more than 50% of its variance [39]. Furthermore, such attempts have overwhelmingly focussed on *Bos taurus*. A key distinction to be made in this field is between *B. taurus* and *B. indicus*, with the latter subspecies generally being better adapted to tropical and arid environments [40]. Consequently, Nellore cattle have relatively good thermal tolerance up until ~35°C [25]. In our study, animal performance was not affected by access to shade under the environmental conditions, but WI and drinking behaviour were clearly affected, suggesting that the animals needed to display some behavioural changes to cope with hot days. Heat waves are becoming more frequent and intense in tropical regions due to global warming. This may affect not only the drinking behaviour of animals but also their performance. Therefore, studies like this should continue in the face of more adverse climate conditions.

## Variability in water intake and drinking behaviour

The substantial individual variability in WI and water-use efficiency among cattle cannot be explained by current models [39]. Some animals have been shown to be more water-efficient, requiring less water per kilogram of BW gained, a trait that is genetically correlated with improved growth performance [39]. Palhares et al. [41] detected the influence of animal performance and productivity on daily WI. This information should be used to propose best practices and support beef systems in regions affected by climate change and water scarcity. In our study, the three top animals ranked by their WI to ADG ratio were 41.7% more efficient (i.e., required less water per kg BW gained) within the +S and 53.8% more efficient within the -S groups (see raw data files [33]) than the three bottom-ranked animals, highlighting the notable individual variability in terms of WI and water use efficiency.

The individual variability can also be explained by the animal-to-animal interactions. Grazing non-lactating dairy cows with the water trough located in the corridor, i.e., with a certain level of restriction of the resource given the distance from the paddock (150 m), showed a greater number of visits per day and time spent drinking for dominant cows in comparison

to subordinate cows [42]. It has been stated that dominance rank in groups of animals may result in a few top-ranking individuals getting plenty and the rest little resources, or the majority getting equal distribution and the lowest ranking animals getting very little [43]. In our study, the variables time spent drinking and visit duration in the -S group had the most variability. This would suggest that social hierarchy may have influenced the use of the limited resource (i.e., the only water trough in the pen) at the peak use times causing difference in the time the steers could spent in the water troughs, where dominant animals made greater use of the resource. The size of the shade was enough to cover all the animals in the pen, so hierarchy of the group would not have affected the use of shade.

Nevertheless, cattle are gregarious and, as such, synchronise their activities [44], which may partially explain the daily pattern and variability in water troughs use. It could be inferred that the animals in -S moved to the water troughs on hot days when the heat stress risk was lower, concentrating the drinking events is fewer hours than those in +S, increasing the dominance of high-ranked animals and increasing variability in drinking behaviours between animals. Additionally, the proximity to the water troughs could itself promote WI, contributing to explain even further the higher WI when shade was not available.

### Practical implication of providing shade to feedlot cattle

The findings of this study emphasise the importance of providing shade as a welfare intervention for cattle in tropical environments [27]. While shade may not consistently improve cattle performance, it effectively reduces the radiant heat load, lowering body-surface and subcutaneous temperatures, respiration rates, and overall WI [9]. For instance, steers in +S showed a reduction of approximately 12 litres in daily WI during the hottest days compared to the coldest (see raw data files [33]). Beyond physiological benefits, shaded environments enhance cattle comfort, reducing heat stress and improving overall welfare, making shade a practical solution in tropical climates [9]. However, the design of shaded areas must be carefully considered to avoid resource disputes [42]. Selective shading for heat-sensitive breeds can optimise costs and enhance welfare, with studies indicating potential payback within four feeding cycles through improved carcass weights [9]. These findings suggest that, despite the upfront cost of providing shade, the long-term benefits, such as water savings, improved performance, and enhanced animal welfare, make it a worthwhile investment [2]. Furthermore, while initial expenses are involved, shade provision offer sustained advantages, with tree planting additionally supporting nutrition, biodiversity, and carbon storage.

### Limitations of the study

The presented study is limited in that the experimental unit within this study was the individual animal, however, individuals were penned in groups. Therefore, it is not possible to entirely rule out group level effects in this study, particularly as cattle are herd animals that do not behave entirely independently of their herd. The issue of pseudoreplication is relatively common across the field of animal science [45–47] and results should be interpreted with consideration of that. Another limiting aspect is that it cannot fully represent the wide variety of factors that potentially impact WI. It also only represents a snapshot of these animals' lives. When considering a greater range and extreme of variables, or studying an animal's entire life, the extent and nature of trends observed are likely to differ, one way or another, from those presented here. A key future area of research would be to analyse the long-term effects of shade provision on cattle performance. This could include the additions of mapping shade availability (based on sun position) at a landscape level, cross-referenced with GPS cattle movement data, WI data, weather measurements, and animal health and performance metrics.

### Conclusions

This study advances the field by providing novel data on the drinking behaviour and individual intake responses of confined beef cattle to artificial shading under tropical conditions, offering incremental yet critical insights into practical

management strategies that enhance animal welfare and productivity in the face of escalating climate challenges. This study demonstrates the substantial impact of environmental factors, particularly shade provision, on the drinking behaviour and thermoregulation strategies of Nellore steers in tropical feedlots. Access to shade altered water intake patterns, allowing steers to drink less frequently but more efficiently per visit, ultimately reducing overall water intake. The findings suggest that cattle with shade adapt by prioritising water intake during cooler times of the day, supporting the idea that shaded environments can effectively help manage heat stress.

Despite variations in drinking behaviour, growth rates and dry matter intake remained similar between shaded and unshaded cattle, evidencing that even under conditions of moderate heat stress Nellore cattle can maintain growth performance without additional cooling interventions. However, the results indicate that while shade did not directly affect growth performance in this study, it served as an essential tool for reducing water intake needs and supporting thermoregulation through behavioural adaptations, which aligns with previous research on the resilience of *B. indicus* breeds in warmer climates.

## Supporting information

**S1 File. Supplement A. Spearman's correlations coefficients and their p-values among all the meteorological variables and the drinking behaviour variables of Nellore steers with and without access to shade in a tropical feedlot.**
(XLSX)

**S2 File. Inclusivity-in-global-research-questionnaire.**
(DOCX)

## Acknowledgments

Rothamsted Research, Embrapa and Andrew S. Cooke are members of the Global Farm Platform initiative (www. globalfarmplatform.org), a global network collaboratively working towards sustainable ruminant livestock production systems.

## Author contributions

**Conceptualization:** Julio C. Pascale Palhares.

**Data curation:** M. Jordana Rivero, Andrew S. Cooke.

**Formal analysis:** M. Jordana Rivero, Andrew S. Cooke.

**Funding acquisition:** M. Jordana Rivero, Julio C. Pascale Palhares.

**Investigation:** M. Jordana Rivero, Julio C. Pascale Palhares, Taisla Inara Novelli, Luciane Silva Martello, Andrew S. Cooke.

**Methodology:** Julio C. Pascale Palhares, Taisla Inara Novelli, Luciane Silva Martello.

**Project administration:** M. Jordana Rivero, Julio C. Pascale Palhares.

**Resources:** Julio C. Pascale Palhares.

**Supervision:** Julio C. Pascale Palhares.

**Visualization:** Andrew S. Cooke.

**Writing – original draft:** M. Jordana Rivero, Simon Perez-Marquez, Andrew S. Cooke.

**Writing – review & editing:** M. Jordana Rivero, Julio C. Pascale Palhares, Taisla Inara Novelli, Luciane Silva Martello, Andrew S. Cooke.

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
