## [Decision Letter · Decision Letter 0]

17 Mar 2025

PONE-D-25-07671Shade provision and its influence on water consumption and drinking behaviour of Nellore cattle in feedlot in a tropical environmentPLOS ONE

Dear Dr. Rivero,

Thank you for submitting your manuscript to PLOS ONE. After careful consideration, we feel that it has merit but does not fully meet PLOS ONE’s publication criteria as it currently stands. Therefore, we invite you to submit a revised version of the manuscript that addresses the points raised during the review process.

We look forward to receiving your revised manuscript.

Kind regards,

Juan J Loor

Academic Editor

PLOS ONE

Journal Requirements:

“This research was partially funded by the Alliance for Sustainable Agriculture (ASA) Partnership award “Measuring Sustainability Metrics for Ruminant Livestock Production Systems in Brazil and UK for a global assessment” funded by the Newton Fund. Rothamsted Research receives strategic funding from the Biotechnological and Biological Sciences Research Council (BBSRC) of the United Kingdom. Support in writing up the work was greatly received by BBSRC through the strategic program Soil to Nutrition (S2N; BBS/E/C/000I0320) and Growing Health (BB/X010953/1). This work was partially supported Coordenaçao de Aperfeiçoamento de Pessoal de Nível Superior – Brazil (CAPES) - Finance Code 001.”

“This research was partially funded by the Alliance for Sustainable Agriculture (ASA) Partnership award “Measuring Sustainability Metrics for Ruminant Livestock Production Systems in Brazil and UK for a global assessment” funded by the Newton Fund. Rothamsted Research receives strategic funding from the Biotechnological and Biological Sciences Research Council (BBSRC) of the United Kingdom. Support in writing up the work was greatly received by BBSRC through the strategic program Soil to Nutrition (S2N; BBS/E/C/000I0320) and Growing Health (BB/X010953/1). This work was partially supported Coordenaçao de Aperfeiçoamento de Pessoal de Nível Superior – Brazil (CAPES) - Finance Code 001.”

6. In the online submission form, you indicated that “Provisionally, the data underlying the results presented in the study are available from Jordana Rivero (jordana.rivero.visiting@rothamsted.ac.uk) .”

7. When completing the data availability statement of the submission form, you indicated that you will make your data available on acceptance. We strongly recommend all authors decide on a data sharing plan before acceptance, as the process can be lengthy and hold up publication timelines. Please note that, though access restrictions are acceptable now, your entire data will need to be made freely accessible if your manuscript is accepted for publication. This policy applies to all data except where public deposition would breach compliance with the protocol approved by your research ethics board. If you are unable to adhere to our open data policy, please kindly revise your statement to explain your reasoning and we will seek the editor's input on an exemption. Please be assured that, once you have provided your new statement, the assessment of your exemption will not hold up the peer review process.

8. We note that you have included the phrase “data not shown” in your manuscript. Unfortunately, this does not meet our data sharing requirements. PLOS does not permit references to inaccessible data. We require that authors provide all relevant data within the paper, Supporting Information files, or in an acceptable, public repository. Please add a citation to support this phrase or upload the data that corresponds with these findings to a stable repository (such as Figshare or Dryad) and provide and URLs, DOIs, or accession numbers that may be used to access these data. Or, if the data are not a core part of the research being presented in your study, we ask that you remove the phrase that refers to these data.

9. Please include your full ethics statement in the ‘Methods’ section of your manuscript file. In your statement, please include the full name of the IRB or ethics committee who approved or waived your study, as well as whether or not you obtained informed written or verbal consent. If consent was waived for your study, please include this information in your statement as well.

Reviewers' comments:

Reviewer's Responses to Questions

**Comments to the Author**

1. Is the manuscript technically sound, and do the data support the conclusions?

Reviewer #1: Partly

2. Has the statistical analysis been performed appropriately and rigorously? 

Reviewer #1: Yes

3. Have the authors made all data underlying the findings in their manuscript fully available?

Reviewer #1: Yes

4. Is the manuscript presented in an intelligible fashion and written in standard English?

Reviewer #1: Yes

5. Review Comments to the Author

Reviewer #1: The authors describe drinking behaviour of beef cattle with/without shade availability, a topic that has not been well researched despite close links with welfare and, therefore, is of high importance considering rising temperatures due to global warming. The article is in general well written. However, there is a lack of consistency in wording that needs to be addressed and authors provide only n=2 per group which is a very small sample size. In many cases it is unclear if values are calculated per pen or per animal, this must be more transparent.

It is described that in heat animals start to play with water instead of drinking – how did you ensure to measure intake or consumption?

For me, the dataset is this small, that I suggest to change the original research article into a short communication.

L17 Please move “steers” before brackets : Nellow Steers (Bos indicus)

L19 rephrase to “allocated in two groups”

L 26 Hard to read, please shorten or split in two sentences: Environmental factors such as Air temperature, humidity, and solar radiation influenced water intake in both groups, with higher air temperatures increasing water intake due to greater drinking frequency and higher relative humidity reducing water intake by decreasing visit frequency.

L29 water demand, not requirements – please pay attention to wording consistency throughout the manuscript

L38 please also indicate the threat of heat stress for animal health and welfare

L42 space lacks at the end of the sentence before bracktes with source

L46 Cattle use evaporative cooling to dissipate heat load. However, evaporative cooling increases the cattle´s need for water to maintain homeostasis [6], consequently

increasing the demand of the production systems.

L59 please delete water before drinking behavior and in the following text. Generally maintain the same phrase (drinking behavior OR water drinking behavior, better drinking behavior; water intake OR water consumption, better water consumption)

L83 months, not mo.

L88 May not an unequal number of animals per group per see affect outcomes as drinking frequencies, behaviors etc ? Did you check your data regarding a potential effect of access per animal?

L98 Please provide, if possible, any information of the wells´ water quality

L103 water consumed instead of drunk, please change through the whole document

How was water consumption measures? Which devices were used, how were the water trough sized?

L119 (“W)

Fig 1: Provide the size of pens in the figure, it would additionally be desirable to have a picture/sizes of the trough/shade constructions.

L129 Was normal distribution of variables checked by wilcox rank test? Please specify.

Please indicate the packages used, also to illustrate the figures

L141 ranged from.. to..

Please use protected spaces before units

L162 was is meant by V= ?

L162 who is “them”? Individuals of this group? Generally, use (cattle OR animals OR individuals and maintain consistency)

L164 What is meant by spending 39% more?time? Here also use protected spaces before ALL units in the whole document.

LL170 p = 0.053 = 0.05 = significant, no tendency needed here.

L174 X and Y correlated positively (r = , p < )

L176 delete “than with shade”

L177 what means 5-6 fold?

L185-193 This information is doubled, as its shown in Fig. 3. However, I understand why mentioning key correlations. You may indicate exact r values at the end of the phrases to provide additional information and/or remove the Fig. in Supplementary File.

L200 specify “affected”

224 highlight, however, that

242 consistent wording- better demand than requirement

243 please delete/rephrase “elsewhere”

252 replace it with shade

Rephrase “water search frequency” the frequency cattle seek for water?

L310 Please indicate a source for this statement

395-397 repetition of “justify investments”

6. PLOS authors have the option to publish the peer review history of their article (what does this mean? ). If published, this will include your full peer review and any attached files.

**Do you want your identity to be public for this peer review?** For information about this choice, including consent withdrawal, please see our Privacy Policy .

Reviewer #1: No

---

## [Author Response · Author response to Decision Letter 1]

14 May 2025

Reviewer’s comments (R)

Authors’ response (A)

R: The authors describe drinking behaviour of beef cattle with/without shade availability, a topic that has not been well researched despite close links with welfare and, therefore, is of high importance considering rising temperatures due to global warming. The article is in general well written.

A: Thank you for your valuable comments and time dedicated to review our manuscript. We have addressed all your comments. A clean version of the revised version highlights the changes in blue text.

R: However, there is a lack of consistency in wording that needs to be addressed and authors provide only n=2 per group which is a very small sample size. In many cases it is unclear if values are calculated per pen or per animal, this must be more transparent.

A: Values are calculated per individual animal. Text has been added to clarify the RFID tags were uses to track individual cattle.

R: It is described that in heat animals start to play with water instead of drinking – how did you ensure to measure intake or consumption?

A: Individual water intake was measured using the Intergado™ System (Intergado Ltd., Contagem, Minas Gerais, Brazil), which employs RFID technology to track individual cattle and water flow meters installed in the troughs to precisely record the volume of water consumed during each drinking event. The system logs only the water volume that flows through the trough during a drinking event, as detected by the flow meters, ensuring that non-consumptive interactions, such as splashing or playing with water, are not recorded as intake. Additionally, the water troughs were designed and tailored to beef cattle drinking behaviour, minimizing spillage and supporting accurate measurement water intake.

R: For me, the dataset is this small, that I suggest changing the original research article into a short communication.

A: The journal does not specifically take short communications, though it can occasionally be a prefix in the title and may be down to editor discretion here. Given that smaller sample sizes are more prone to Type II errors, the extent to which significant differences and associations have been found in our study is, we believe, relatively convincing, especially given the little research done in this area. We would also note that the sample size is higher than numerous other livestock behaviour full papers in this journal:

N= 20 (10.1371/journal.pone.0285933)

N = 19 (10.1371/journal.pone.0131632)

N = 15 (10.1371/journal.pone.0265037)

N = 24 (10.1371/journal.pone.0118617)

N = 18 (10.1371/journal.pone.0144583)

R: L17 Please move “steers” before brackets : Nellow Steers (Bos indicus)

A: Done.

R: L19 rephrase to “allocated in two groups”

A: Done.

R: L 26 Hard to read, please shorten or split in two sentences: Environmental factors such as Air temperature, humidity, and solar radiation influenced water intake in both groups, with higher air temperatures increasing water intake due to greater drinking frequency and higher relative humidity reducing water intake by decreasing visit frequency.

A: Rephrased and split.

R: L29 water demand, not requirements – please pay attention to wording consistency throughout the manuscript

A: The expression “water demand” was used consistently throughout the document.

R: L38 please also indicate the threat of heat stress for animal health and welfare

A: Added.

R: L42 space lacks at the end of the sentence before bracktes with source

A: Space added.

R: L46 Cattle use evaporative cooling to dissipate heat load. However, evaporative cooling increases the cattle´s need for water to maintain homeostasis [6], consequently increasing the demand of the production systems.

A: Rephrased.

R: L59 please delete water before drinking behavior and in the following text. Generally maintain the same phrase (drinking behavior OR water drinking behavior, better drinking behavior; water intake OR water consumption, better water consumption) “drinking behaviour” used throughout the manuscript.

A: We prefer to keep “water intake” consistently throughout the document since this is how the response variable (WI) was defined and presented in tables and graphs.

R: L83 months, not mo.

A: Changed.

R: L88 May not an unequal number of animals per group per see affect outcomes as drinking frequencies, behaviors etc ? Did you check your data regarding a potential effect of access per animal?

A: Group sizes were balanced except for one pen, which had one fewer animal due to an illness after the adaptation period, with no replacement available. While we acknowledge the theoretical possibility that group size differences could affect access to resources and influence drinking behaviours, this was not observed in our study. The water troughs, designed and sized by Intergado Ltd. according to manufacturer specifications for beef cattle, provided adequate capacity and access for all animals in each pen, minimising competition. The Intergado™ System recorded consistent metrics across groups. Statistical analysis of these variables showed no significant differences attributable to the minor variation in group size, confirming that access to the troughs was not restricted and that drinking frequencies and behaviours remained unaffected.

R: L98 Please provide, if possible, any information of the wells´ water quality

A: There is no information available.

R: L103 water consumed instead of drunk, please change through the whole document

A: “drunk” replaced throughout the manuscript.

R: How was water consumption measures? Which devices were used, how were the water trough sized?

A: In the original manuscript, we briefly mentioned the use of the Intergado™ System (Intergado Ltd., Contagem, Minas Gerais, Brazil) for monitoring individual water intake, as referenced in Chizzotti et al. (2015).

The Intergado™ System utilises RFID technology to identify individual animals and water flow meters attached to the troughs to measure the volume of water consumed during each drinking event. The troughs were designed and sized by Intergado Ltd. to accommodate the cattle’s drinking behaviour, ensuring sufficient access and capacity for accurate measurement, though specific dimensions were not detailed in our study as they followed the manufacturer’s standard specifications for beef cattle. We have included this new information in the manuscript. Similarly, we enhanced the description of the feed intake masurement.

R: L119 (“W)

A: Amended.

R: Fig 1: Provide the size of pens in the figure, it would additionally be desirable to have a picture/sizes of the trough/shade constructions.

A: As mentioned in the manuscript, each pen is 400 m2. We have added photos of all the relevant features in the pen.

R: L129 Was normal distribution of variables checked by wilcox rank test? Please specify. Please indicate the packages used, also to illustrate the figures

A: The Wilcox rank test does not require a normal distribution and thus testing is not necessary. The decision to use non-parametric tests for this was a factor of both sample size and non-normal distributions of variables and the desire for consistency in testing methodologies. Text has been added in this regard. We have also changed the model used for the data to a zero inflated model to better account for the distribution of the data. Citations for R packages added.

R: L141 ranged from.. to.. Please use protected spaces before units

A: Amended

R: L162 was is meant by V= ?

A: This is Wilcoxon’s V, a test statistic. It represents the sum of the ranks of positive differences between the treatment groups.

R: L162 who is “them”? Individuals of this group? Generally, use (cattle OR animals OR individuals and maintain consistency)

A: Amended

R: L164 What is meant by spending 39% more?time? Here also use protected spaces before ALL units in the whole document.

A: Sentence slightly amended. The expression “spending xx% more time” means the duration (e.g., in minutes or seconds) was greater by that percentage compared to the other group.

R: L170 p = 0.053 = 0.05 = significant, no tendency needed here.

A: We respectfully disagree with the reviewer. The limit for significance is 0.05. Only values up to 0.050 are considered significant. Hence, given that 0.053 > 0.050, this is a trend only.

R: L174 X and Y correlated positively (r = , p < )

A: Changed.

R: L176 delete “than with shade”

A: Done

R: L177 what means 5-6 fold?

A: The term "5-6 fold" means an increase of 5 to 6 times the original value (i.e., a multiplication factor of 5 to 6). This phrase is a common scientific shorthand.

R: L185-193 This information is doubled, as its shown in Fig. 3. However, I understand why mentioning key correlations. You may indicate exact r values at the end of the phrases to provide additional information and/or remove the Fig. in Supplementary File.

A: Here we are describing the results in terms of correlations (as we do with tables). We agree that the exact r-value should be provided and this has now been added for the significant correlations highlight However, we prefer to maintain the figure in the main document as this provides additional information. A supplementary table with the exact r and p values has been added.

R: L200 specify “affected”

A: Changed to ‘associated’

R: 224 highlight, however, that

A: We disagree with the suggestion. The phrase is correct in grammar and meaning.

R: 242 consistent wording- better demand than requirement

A: Done throughout the whole manuscript.

R: 243 please delete/rephrase “elsewhere”

A: Done.

R: 252 replace it with shade

A: Done

R: L256 Rephrase “water search frequency” the frequency cattle seek for water?

A: Rephrased to enhance clarity, but the expression “water search frequency” was retained as it’s grammatically correct.

R: L310 Please indicate a source for this statement

A: Added.

R: 395-397 repetition of “justify investments”

A: Sentences rephrased.

---

## [Decision Letter · Decision Letter 1]

27 Jun 2025

PONE-D-25-07671R1Shade provision and its influence on water intake and drinking behaviour of Nellore cattle in feedlot in a tropical environmentPLOS ONE

Dear Dr. Rivero,

Thank you for submitting your manuscript to PLOS ONE. After careful consideration, we feel that it has merit but does not fully meet PLOS ONE’s publication criteria as it currently stands. Therefore, we invite you to submit a revised version of the manuscript that addresses the points raised during the review process.

We look forward to receiving your revised manuscript.

Kind regards,

Juan J Loor

Academic Editor

PLOS ONE

**Additional Editor Comments:**

I COMMEND THE AUTHORS FOR THE EFFORT IN ADDRESSING THE COMMENTS FROM THE ORIGINAL REVIEWER. UNFORTUNATELY, THE REVIEWER DECLINED TO REVIEW THE REVISED PAPER AND GIVEN MY LACK OF EXPERTISE IN THIS AREA OF RESEARCH I SOUGHT ADDITIONAL REVIEWS FROM TWO INDIVIDUALS WITH THE REQUIRED EXPERTISE. ALTHOUGH ONE FO THE REVIEWERS BELEIVES THERE IS A LACK OF INNOVATIVE ASPECTS IN THE RESEARCH, I DO NOT BELIEVE THAT IN-AND-OF ITSELF IS A REASON TO REJECT THE PAPER. PLEASE ADDRESS THIS ASPECT BY PROVIDING GREATER CONTEXT AND, PERHAPS, HOW THIS WORK MOVES THE FIELD FORWARD ALBEIT IN AN INCREMENTAL WAY. LASTLY, PLEASE ADDRESS THE SPECIFIC QUESTIONS POSED BY THE SECOND REVIEWER AS WELL. ENSURE THAT THE REVISED MANUSCRIPT REFLECTS THE CHANGES MADE TO ADDRESS BOTH OF THE REVIEWERS.

Reviewers' comments:

Reviewer's Responses to Questions

**Comments to the Author**

1. If the authors have adequately addressed your comments raised in a previous round of review and you feel that this manuscript is now acceptable for publication, you may indicate that here to bypass the “Comments to the Author” section, enter your conflict of interest statement in the “Confidential to Editor” section, and submit your "Accept" recommendation.

Reviewer #2: (No Response)

Reviewer #3: (No Response)

2. Is the manuscript technically sound, and do the data support the conclusions?

Reviewer #2: No

Reviewer #3: Yes

3. Has the statistical analysis been performed appropriately and rigorously? 

Reviewer #2: No

Reviewer #3: Yes

4. Have the authors made all data underlying the findings in their manuscript fully available?

Reviewer #2: Yes

Reviewer #3: Yes

5. Is the manuscript presented in an intelligible fashion and written in standard English?

Reviewer #2: No

Reviewer #3: Yes

6. Review Comments to the Author

Reviewer #2: L84: Use body weight (BW) instead of liveweight (LW) for consistency and clarity.

L91–94: Why was the initial BW estimated using a regression based on only two time points? This approach is typically used for calculating average daily gain (ADG) when three or more data points are available. Alternatively, consider using the average BW from two consecutive weighing days.

L96: Were animals fed ad libitum or were they restricted to a specific amount? Please clarify this in the text.

L98–100: What was the nutrient composition of the diet? Include a table with detailed dietary composition.

L104–106: Please rephrase this sentence for clarity and flow.

L123–126 (Fig. 1): Why were the +S pens all located on the same side (east) of the facility? Was there no randomization in pen assignment?

L136–137: I strongly disagree with the approach taken here. While individual animal data can be collected for dry matter intake (DMI) and water intake (WI), the pen should be considered the experimental unit when comparing treatments. The shade treatment was applied at the pen level, not at the individual animal level, and thus treatment effects cannot be independently assessed at the animal level.

Reviewer #3: I recommend rejection of this manuscript due to a lack of innovation. Although the study was well designed and the statistical analyses are appropriate, the manuscript does not offer any new or relevant insight into the topic under investigation. The discussion is well grounded in the literature, showing that what they have discovered can easily be found in the literature. But overall, I don't believe the work has the level of impact required for publication in this journal.

As the authors themselves recognize, future studies could explore the long-term effects of shade provision on cattle performance using a more integrated approach. For example, incorporating landscape-level shade mapping (based on sun position), GPS tracking of cattle movements, water intake, weather conditions and animal health and performance metrics would result in a more comprehensive and impactful study. I encourage authors to consider expanding the scope of their research in this direction to strengthen future submissions.

7. PLOS authors have the option to publish the peer review history of their article (what does this mean? ). If published, this will include your full peer review and any attached files.

**Do you want your identity to be public for this peer review?** For information about this choice, including consent withdrawal, please see our Privacy Policy .

Reviewer #2: No

Reviewer #3: No

---

## [Author Response · Author response to Decision Letter 2]

1 Jul 2025

Dear Reviewers,

Thank you for reviewing our manuscript. We have addressed your comments in the Revision Note and modified the manuscript accordingly.

We hope you find our responses and modifications satisfactory.

Thank you again.

Sincerely,

Jordana Rivero

---

## [Decision Letter · Decision Letter 2]

5 Aug 2025

PONE-D-25-07671R2Shade provision and its influence on water intake and drinking behaviour of Nellore cattle in feedlot in a tropical environmentPLOS ONE

Dear Dr. Rivero,

Thank you for submitting your manuscript to PLOS ONE. After careful consideration, we feel that it has merit but does not fully meet PLOS ONE’s publication criteria as it currently stands. Therefore, we invite you to submit a revised version of the manuscript that addresses the points raised during the review process.

We look forward to receiving your revised manuscript.

Kind regards,

Juan J Loor

Academic Editor

PLOS ONE

Journal Requirements:

Reviewers' comments:

Reviewer's Responses to Questions

**Comments to the Author**

1. If the authors have adequately addressed your comments raised in a previous round of review and you feel that this manuscript is now acceptable for publication, you may indicate that here to bypass the “Comments to the Author” section, enter your conflict of interest statement in the “Confidential to Editor” section, and submit your "Accept" recommendation.

Reviewer #3: All comments have been addressed

2. Is the manuscript technically sound, and do the data support the conclusions?

Reviewer #3: Partly

3. Has the statistical analysis been performed appropriately and rigorously? 

Reviewer #3: Yes

4. Have the authors made all data underlying the findings in their manuscript fully available?

Reviewer #3: Yes

5. Is the manuscript presented in an intelligible fashion and written in standard English?

Reviewer #3: Yes

6. Review Comments to the Author

Reviewer #3: 1: Should environmental variables be considered part of the Results or Methods section? In my opinion, the THI within the pen should be treated as a result, since it was affected by the treatment (shade). However, the other environmental variables (e.g., ambient temperature, rainfall, etc.) should remain in the Methods section, as they are not controlled by the experiment and are more appropriately described as background conditions.

2: I suggest presenting the diet ingredients and nutrient composition in a separate table rather than in the text.

3: This is just a general comment for consideration in future studies, no changes to the current manuscript are needed because de explanation is already in the limitations. While I understand and agree that the individual tracking system (feed and water troughs) allowed for the measurement of individual-level responses, I respectfully disagree with the classification of the individual animal as the experimental unit. This issue led to a lengthy discussion "Variability in water intake and drinking behaviour"

From a statistical and conceptual standpoint, the experimental unit is defined as the smallest unit to which a treatment is applied. In this case, the shade treatment was pen.

7. PLOS authors have the option to publish the peer review history of their article (what does this mean? ). If published, this will include your full peer review and any attached files.

**Do you want your identity to be public for this peer review?** For information about this choice, including consent withdrawal, please see our Privacy Policy .

Reviewer #3: No

---

## [Author Response · Author response to Decision Letter 3]

5 Aug 2025

Thank you for your input. Responses uploaded as an attachment.

---

## [Editor Report · Decision Letter 3]

13 Aug 2025

Shade provision and its influence on water intake and drinking behaviour of Nellore cattle in feedlot in a tropical environment

PONE-D-25-07671R3

Dear Dr. Rivero,

We’re pleased to inform you that your manuscript has been judged scientifically suitable for publication and will be formally accepted for publication once it meets all outstanding technical requirements.

Kind regards,

Juan J Loor

Academic Editor

PLOS ONE
---

## [Editor Report · Acceptance letter]

PONE-D-25-07671R3

PLOS ONE

Dear Dr. Rivero,

I'm pleased to inform you that your manuscript has been deemed suitable for publication in PLOS ONE. Congratulations! Your manuscript is now being handed over to our production team.

Kind regards,

on behalf of

Dr. Juan J Loor

Academic Editor

PLOS ONE